# Gas-Phase TiO$_2$ Photosensitized Mineralization of Some VOCs: Mechanistic Suggestions through a Langmuir–Hinshelwood Kinetic Approach

**Marta Bettoni [1], Stefano Falcinelli [1],\*, Cesare Rol [2], Marzio Rosi [1] and Giovanni Vittorio Sebastiani [1]**

[1] Dipartimento di Ingegneria Civile ed Ambientale, Università di Perugia, Via G. Duranti 93, 06125 Perugia, Italy; marta.bettoni.mb@gmail.com (M.B.); marzio.rosi@unipg.it (M.R.); gvsebastiani@gmail.com (G.V.S.)

[2] Dipartimento di Chimica, Biologia e Biotecnologie, Università di Perugia, Via Elce di Sotto 8, 06123 Perugia, Italy; cesarepianista@gmail.com

\* Correspondence: stefano.falcinelli@unipg.it; Tel.: +39-075-585-3862

**Abstract:** A jointed experimental and theoretical investigation pointing out new insights about the microscopic mechanism of the volatile organic compounds (VOCs) photocatalytic elimination by TiO$_2$ was done. Methane, hexane, isooctane, acetone and methanol were photomineralized in a batch reactor. Values of K (adsorption constant on TiO$_2$) and k (mineralization rate constant) of the five VOCs (treating the kinetic data through a Langmuir–Hinshelwood approach) were determined. Recorded K (in the range of $0.74 \times 10^{-2}$–$1.11 \times 10^{-2}$ ppm$^{-1}$) and k (in the range of 1.9–9.9 ppm min$^{-1}$) values and performed theoretical calculations allowed us to suggest the involvement of an electron transfer step between the VOC and the hole, TiO$_2$(h$^+$), as the rate-determining one.

**Keywords:** titanium dioxide; oxidation; photocatalysis; VOCs mineralization; Langmuir–Hinshelwood





## 1. Introduction

The TiO$_2$ photosensitized mineralization of most volatile organic compounds (VOCs) as pollutants has been widely considered [1–9] principally due to its biological and chemical inertness, strong oxidizing power, low cost and long-time stability against photo- and chemical corrosion of this semiconductor.

The photocatalytic process is known to involve the VOC pre-adsorption at the sensitizer surface followed by the generation of hole/electron pairs into the semiconductor through the absorption of light with energy equal to or higher than the band-gap energy; the electrons reduce the atmospheric oxygen while the holes oxidize the VOC, directly (through an electron transfer process from the organic substrate to the hole) or indirectly (through the intervention of radicals OH$^\bullet$ derived from the oxidation of adsorbed water by the hole) [10,11].

In current literature, many authors have reported the effect of TiO$_2$ properties on the gas-phase photocatalytic efficiency, while a minor number of papers concern the effect of the VOC structure on the preadsorption at TiO$_2$ surface and on the reaction efficiency. In particular, the considered VOCs have been principally alkanes [12–17], chloroalkanes [12,18], chloroalkenes, [10,12,18], carbonyl derivatives [10,12,15,16,19], alcohols [10,12,18,19], ethers [12] and aromatic compounds [10,12].

Within the previous organic substrates, in this paper, we selected five VOCs (methane, hexane, isooctane, acetone and methanol) that, in our experimental conditions, are quantitatively mineralized to CO$_2$ and H$_2$O. In particular, the experiments were carried out in a batch reactor (irradiation with high-power external lamp through a Pyrex cap).

Preliminarily, in the first topic of this work (see Section 2.1), a series of different TiO$_2$ powders, prepared by a sol–gel method and annealed at different temperatures,

were considered. These powders show properties (such as crystalline form, crystallinity degree, crystal size, surface area, pore volume) that gradually change with the calcination temperature as demonstrated by previous investigations [20] using Brunauer, Emmett and Teller theory (BET), X-ray powder diffraction analysis (XRD) and thermogravimetric analysis (TG). These samples were used to verify if the relative efficiency of two substrates with different polarity (hexane and acetone) is maintained, changing the type of $TiO_2$ used. In particular, in this work, the different $TiO_2$ powders were used to realize a profile "$TiO_2$ efficiency vs. powder annealing temperature" in the presence of hexane as a nonpolar substrate at 600 ppm substrate concentration (the same experimental conditions used in a previous study on acetone [20] in order to make a comparison possible), evaluating the rate through a simplified Langmuir–Hinshelwood (LH) treatment, as the apparent rate ($k_a$, as discussed in Equation (3), Section 2.1); these data were compared with the same measurements previously performed for acetone as a polar substrate [20].

A second topic (see Section 2.2) that was considered in this work involves the determination of the values of K and k for the above five substrates in the presence of $TiO_2$ P25, a very reactive commercial powder (see Section 3.1), changing the initial VOC concentration and treating the kinetic data with LH equation (see Equation (4) and Section 2.2).

The presented data allowed us to obtain important mechanistic information about the considered substrates oxidation reactions. A careful analysis of the kinetic experimental evidence together with theoretical considerations was done (see below) in order to clarify if the rate-determining step in the reaction mechanism of the investigated photosensitive oxidations is an electron transfer step ($RH \rightarrow RH^+ + e^-$) instead of a possible competitive homolytic process ($RH \rightarrow R^\bullet + H^\bullet$). To obtain this information, which constitutes the novelty of our work, a jointed experimental and theoretical investigation was done: (i) chemical kinetics experiments via an LH approach for the determination of either K (the adsorption constant on $TiO_2$) and k (the mineralization rate constant) using five VOCs (methanol, acetone, isooctane, hexane, and methane) were performed; (ii) computational analysis through the optimization of the geometry of the involved neutral and the corresponding radical cations species was done and the ionization potentials of the considered substrates, both in the absence and in the presence of water molecules, always present at the semiconductor surface (see Section 2.3), were determined. To our knowledge, there are no mechanistic studies concerning photocatalytic $TiO_2$ mineralization of VOCs in gaseous phase in the literature. The present work wants to make a contribution in this sense but does not claim to carry out a study of the complex reaction mechanism of the investigated processes, but only to point out the characteristics of the rate-determining step using the mathematical LH approach. This type of kinetic experiments coupled with theoretical investigations performed with a high level of accuracy has important applications in the development of innovative photocatalytic reactors for the treatment of gaseous environmental pollutants and for industrial purposes.

## 2. Results and Discussion

### 2.1. Photooxidation of Hexane with Different Synthetic $TiO_2$ Powders

It is useful to discuss as first the topic relative to the data of photocatalytic efficiency obtained in the presence of different synthetic powders, considering hexane (a nonpolar VOC) as substrate; this allows to compare the behavior of this substrate with that of a previously studied VOC, acetone (a polar substrate) [20].

As previously reported [20,21], the considered powders were prepared by hydrolysis of TTIP and by subsequent calcination at different temperatures. In this way, various powders with different properties, such as crystalline form (anatase and/or rutile), crystallinity degree, crystal size, surface area and pore volume [20,21], were synthesized. The discussion relative to the change of each of the above properties as a function of the calcination temperature of the different powders were reported in previous articles [20,21].

The data relative to the reaction stoichiometry, as residual hexane and produced $CO_2$, are reported in Section 2.2 and support the complete mineralization of the substrate. The

most used equation that allows obtaining the adsorption equilibrium constant (K) and the reaction rate constant (k) from kinetic data is the LH equation (Section 2.1) [3,5].

The hexane photocatalytic degradation follows the LH equation [5] (see Equation (1) below, where r is the rate, k is the true rate constant, K is the adsorption constant of the substrate on TiO2, and C is the substrate concentration).

$$r = kKC/(1 + KC) \qquad (1)$$

As the considered initial concentration of hexane (600 ppm: the same concentration value previously used in our laboratory in the case of similar kinetic experiments involving acetone [20]) is low, we have utilized a simplified LH equation (see Equation (2) below) where $k_a$, the apparent rate constant, can represent a quantitative kinetic measure of the powder relative efficiency (see, for example, ref. [22]).

$$r = kKC = k_aC \qquad (2)$$

The integrated form of the rate equation, as the function of the time t, is reported in Equation (3), where $C_0$ is the substrate starting concentration.

$$\ln(C_0/C) = \ln[100/(100 - \%\,CO_2)] = k_a t \qquad (3)$$

In this way, from the plots in Figure 1, we have obtained a series of kinetic linear correlations for the different synthetic powders (see Table 1, where R is the correlation coefficient).

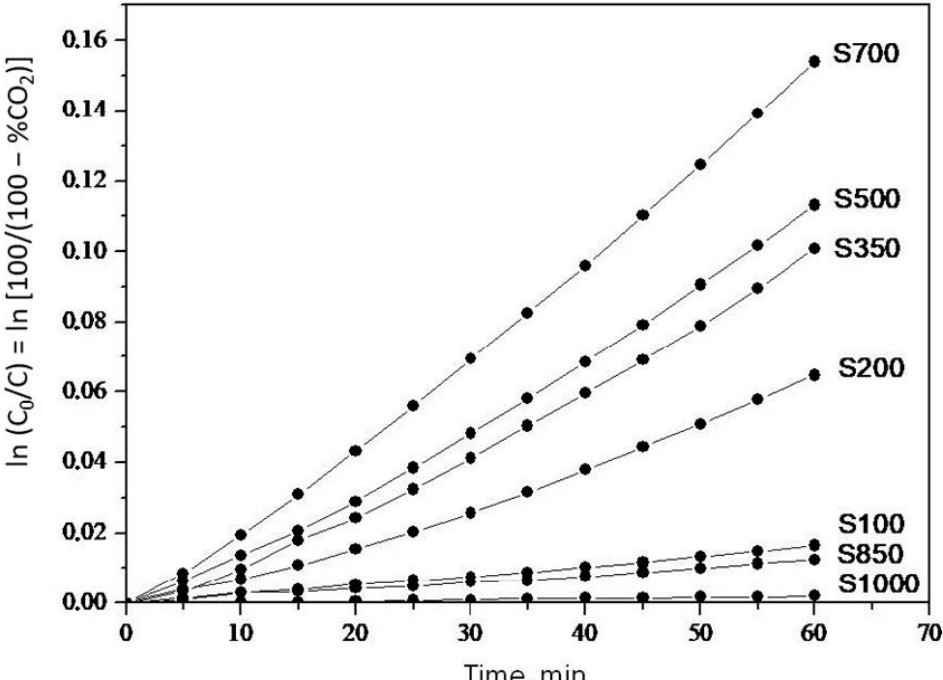

**Figure 1.** Langmuir–Hinshelwood kinetic plots (see Equation (3)) based on $CO_2$ produced from the photooxidation of hexane (600 ppm) sensitized by synthetic $TiO_2$ samples (calcined from 100 °C, S100, to 1000 °C, S1000). The experimental error is ≤10%.

**Table 1.** Apparent rate constants ($k_a$) from the Langmuir–Hinshelwood treatment (see Equation (3)) based on $CO_2$ produced from the $TiO_2$ samples photosensitized oxidation of hexane (concentration: 600 ppm; reaction time: 1 h). R is the correlation coefficient.

| Sample | $k_a \times 10^3$, min$^{-1}$ | R |
| --- | --- | --- |
| S100 | 0.260 | 0.9988 |
| S200 | 0.990 | 0.9907 |
| S350 | 1.500 | 0.9916 |
| S500 | 1.700 | 0.9947 |
| S700 | 2.400 | 0.9973 |
| S850 | 0.200 | 0.9972 |
| S1000 | 0.032 | 0.9908 |

In Figure 2 are reported the values of the slopes ($k_a$) obtained until one hour of reaction (starting from 600 ppm of hexane and at $12 \pm 3\%$ relative humidity (RH) vs. the $TiO_2$ annealing temperatures.

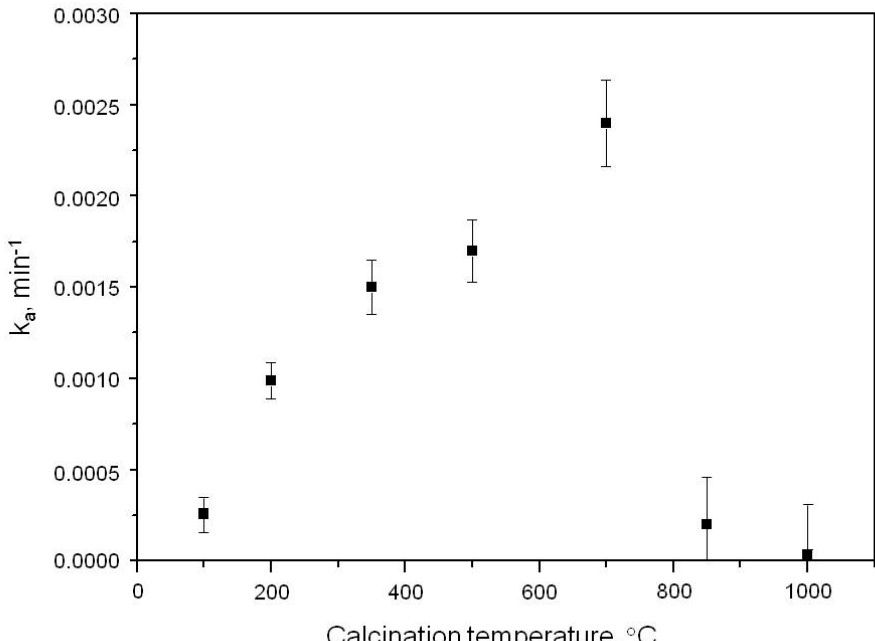

**Figure 2.** The dependence of the apparent kinetic constants ($k_a$ in Equation (3)) from the calcination temperature of $TiO_2$ powders.

As observed with acetone [20], the efficiency increases (ca. 10 times) from 100 °C up to 700 °C in the hexane profile (see Figure 2), and this behavior could be attributed to the crystallinity increase of the anatase structure. Otherwise, for both substrates, the efficiency decrease (ca. 20 times) from 700 °C to 1000 °C could be ascribed to a gradual change of the crystal form from anatase (more reactive) to rutile (less reactive). A first observation is that the selection of the most efficient $TiO_2$ (calcinated at 700 °C for both substrates) and the less efficient one (calcinated at 1000 °C for both substrates) is independent of the substrate polarity. Moreover, the relative efficiency of the two substrates is maintained independently from the type of selected $TiO_2$; in fact, acetone is ca. 2–3 times more reactive than hexane with all the synthesized $TiO_2$ powders.

*2.2. Photooxidation of Methane, Hexane, Isooctane, Acetone and Methanol with TiO$_2$ P25*

In this paragraph, we consider the LH equation (Equation (1)) in the form of Equation (4) below, where $r_O$ is the starting rate and $C_O$ is the starting concentration (see, for example, ref. [10]), to obtain the values of k and K of the considered substrates.

$$1/r_o = 1/kK \; 1/C_o + 1/k \qquad (4)$$

Preliminarily, the reaction stoichiometry for the considered substrates was verified by us for acetone [20], for hexane in this work (after 1 h reaction, starting from 600 ppm of hexane and at 12 $\pm$ 3% RH, 250 ppm of hexane residue and 1920 ppm of $CO_2$ are recovered) or on the base of the literature [13–15,18,23,24].

Another preliminary observation is that the used cap material (Pyrex) almost completely avoids the direct photolysis of the substrate, as verified by blank experiments performed in the absence of TiO$_2$. It must be observed that the use of Pyrex as a light filter is an experimental reaction condition, which is crucial to study exclusively the TiO$_2$ sensitized photooxidation mechanism, even if this comes at the expense of substrates conversion. In particular, the less reactive substrate (methane) cannot be quantitatively studied in these experimental conditions: in fact, starting from 600 ppm of the substrate, 23, 35 and 47 ppm of $CO_2$ are obtained, after 1, 2 and 3 h, respectively, that represent a very little amount of $CO_2$, comparable with the results obtained in the absence of TiO$_2$ (7, 13 and 22 ppm).

As shown in Figures 3–6, with each substrate (hexane, isooctane, acetone and methanol), the slope (r from the linear correlation between [$CO_2$] and t) remains nearly constant until 1 h of reaction (R $\geq$ 0.99) for each considered $C_O$ and therefore $r_O$ can be identified with r. In the photocatalytic degradation kinetic data of Figures 3–6, the use of slightly different initial VOCs concentrations is due to the difficulty of manually sampling the same concentration value by the operator.

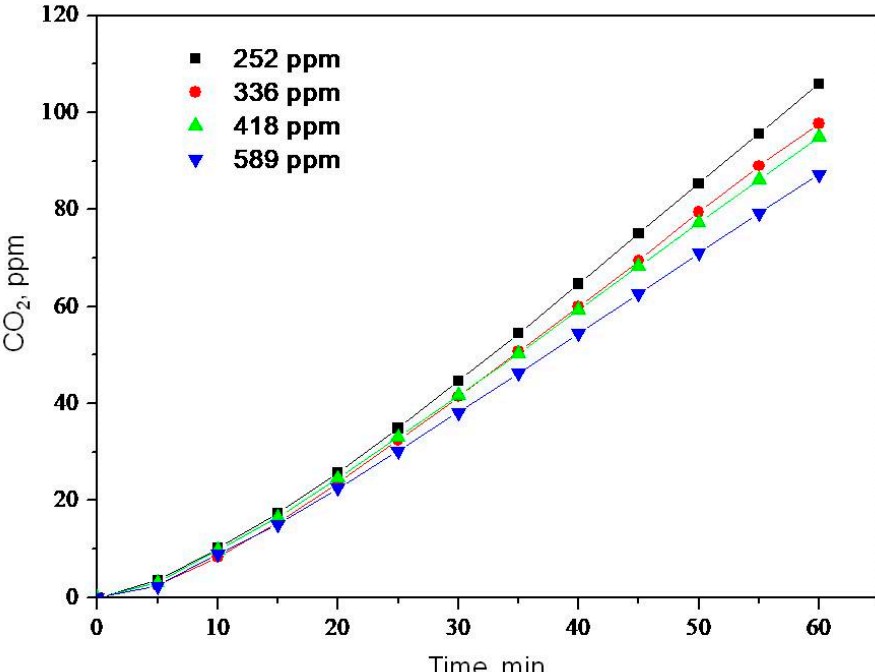

**Figure 3.** Photocatalytic degradation of hexane (as $CO_2$ produced) at different initial concentrations of the substrate. TiO$_2$ (P25, Degussa) commercial powder is used. The experimental error is $\leq$10%.

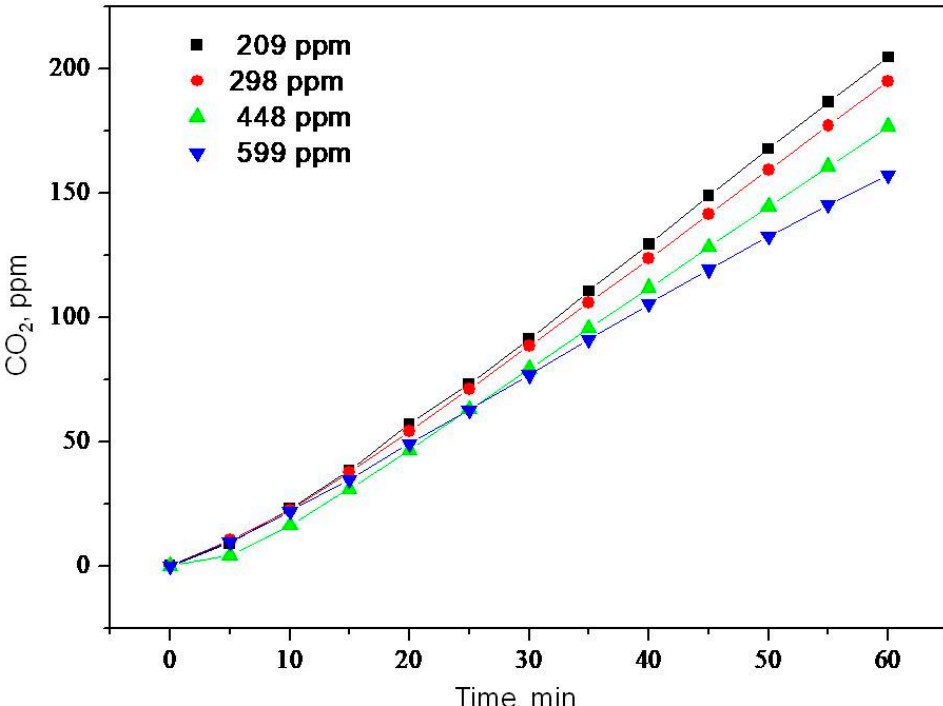

**Figure 4.** Photocatalytic degradation of acetone (as $CO_2$ produced) determined at different initial concentrations of the substrate. $TiO_2$ (P25, Degussa) commercial powder is used. The experimental error is $\leq 10\%$.

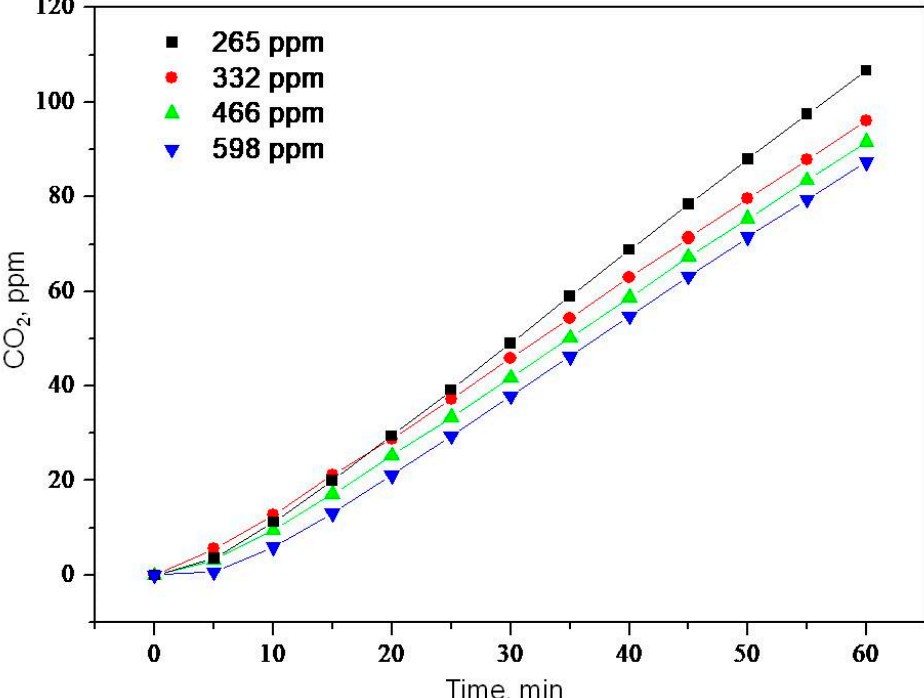

**Figure 5.** Photocatalytic degradation of isooctane (as $CO_2$ produced) determined at different initial concentrations of the substrate. $TiO_2$ (P25, Degussa) commercial powder is used. The experimental error is $\leq 10\%$.

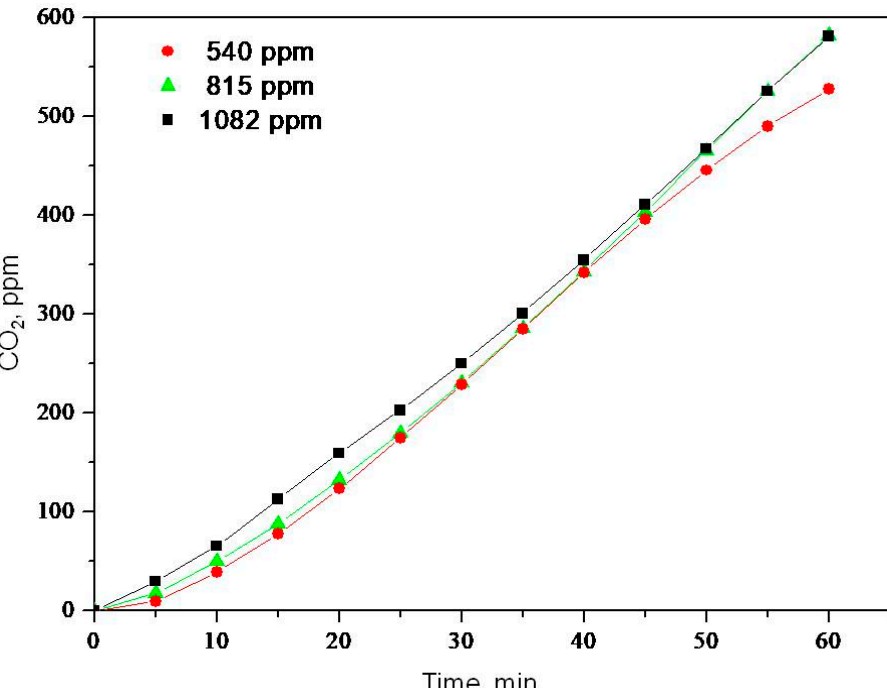

**Figure 6.** Photocatalytic degradation of methanol (as $CO_2$ produced) determined at different initial concentrations of the substrate. $TiO_2$ (P25, Degussa) commercial powder is used. In this case, only three initial substrate concentration values were analyzed owing to the difficulties encountered in manual sampling due to the very high vapor pressure of the methanol. The experimental error is $\leq 10\%$.

In the kinetic data reported in Figure 6, since the photocatalytic degradation of methanol is characterized by the higher reaction rate compared to the other VOCs studied, as shown in Table 2, the final $CO_2$ concentrations detected are considerably higher than in the other cases.

**Table 2.** Langmuir–Hinshelwood parameters (k and K) relative to the photosensitized oxidation of hexane, acetone, isooctane and methanol.

| Substrate | k, ppm min$^{-1}$ | K $\times 10^2$, ppm$^{-1}$ | kK $\times 10^2$, min$^{-1}$ |
|---|---|---|---|
| Hexane | 1.9 | 1.05 | 2.0 |
| Acetone | 3.8 | 1.05 | 4.0 |
| Isooctane | 2.1 | 0.74 | 1.5 |
| Methanol | 9.9 | 1.11 | 11.0 |

In Figure 7 are reported the plots (relative to the four substrates) of $1/r_\bigcirc$ vs. $1/C_\bigcirc$ that allow obtaining the values of k and K for each substrate. The values of the constants are reported in Table 2.

Before discussing the results in Table 2, it must be noted that other authors have determined, through the LH treatment, k and K values relative to, at most, two substrates, but to our knowledge, no author has already used the comparison of these data to deduce mechanistic information [10,12,14]. In fact, the experimental conditions used in these works do not allow to suggest a mechanism exclusively related to the $TiO_2$ photosensitized oxidation as in these conditions the substrates also undergo competitive direct photolysis [10,14] or thermal reaction [12].

As concerns the K values in Table 2, it is evident that they are similar to a significant contribution probably given by Van der Waals (VdW) interactions. As a confirmation, between the two alkanes (hexane and isooctane), the one with a lower contact surface (isooctane) is slightly less adsorbed (lower K).

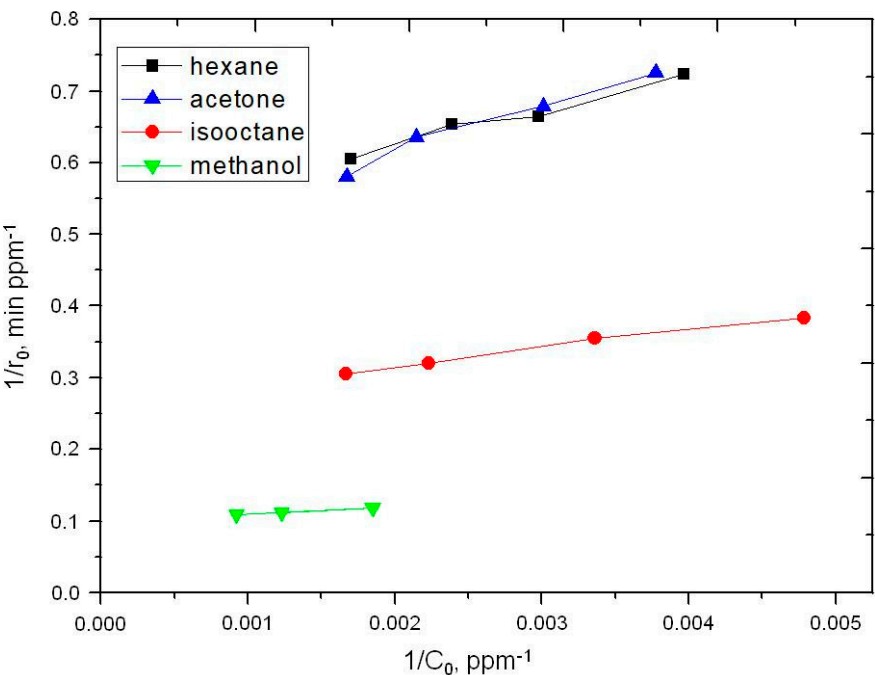

**Figure 7.** Langmuir–Hinshelwood plots (Equation (4)) relative to hexane (square), isooctane (triangle up), acetone (circle) and methanol (triangle down). The experimental error is $\leq 10\%$.

To tentatively evaluate also the relative efficiency of methane (not inserted in Table 2 due to its low reaction rate) it can be considered the amount of $CO_2$ produced at the same reaction time (for example 1 h) from the same amount of substrate (for example 600 ppm of methane and hexane). In this way, methane (see above data in this paragraph) is at least four times less efficient than n-hexane (see Figure 3). The undetectable reactivity (Kk) of methane (not present in Table 2), which presents a minimal contact surface (since a lower VdW interaction), suggests a significant contribution of a very low K to the Kk (efficiency) value of this hydrocarbon.

As concerns the two polar compounds, acetone and methanol, the K values are similar to that of hexane, a nonpolar compound, probably because the minor contribution of the VdW interactions (due to a lower contact surface) could be compensated by a dipole-dipole interaction, for acetone, and dipole-dipole interaction via a hydrogen bond, for methanol, with water always present at the $TiO_2$ surface. As for the k values in Table 2, it can be observed that the nonpolar substrates, hexane and isooctane, are less reactive than the polar ones, acetone and methanol, suggesting an electron transfer rate-determining step, $RH + TiO_2(h^+) \rightarrow RH^+ + TiO_2$, in the reaction mechanism, as observed in the liquid phase (see references cited in [20]). In fact, the radical cations deriving from the polar compounds are more stable than those from the nonpolar ones. The exclusion of a homolytic process, $RH + OH^\bullet \rightarrow R^\bullet + H_2O$, as a rate-determining step, is also supported by the nearly same reactivity of hexane and isooctane. In fact, considering the relative stability of the correspondingly produced radicals in the above step (the tertiary radical obtained from a hydrogen extraction from isooctane is more stable than anyone formed from hexane), isooctane should be more reactive than hexane.

### 2.3. Evaluation of Ionization Potentials

In order to evaluate the operation of an electron transfer mechanism, the ionization potentials (IPs) of methanol, acetone, n-hexane and methane were calculated, optimizing the geometry of the neutral molecules and of the corresponding positive ions (that is, the corresponding radical cations) at B3LYP (Becke, 3-parameter, Lee–Yang–Parr)/aug-cc-pVTZ (correlation-consistent polarized Valence Triple-zeta) level. At the optimized

structures, we performed a single point calculation at Coupled Cluster Single Double (Triple) (CCSD(T))/aug-cc-pVTZ. level in order to have a more accurate energy value. The computed IPs are reported in Table 3, together with the experimental values [25].

**Table 3.** Ionization potential (eV) computed at the CCSD(T)/aug-cc-pVTZ//B3LYP/aug-cc-pVTZ level of theory at 298.15 K for the investigated molecules. Experimental values [25] are reported for comparison.

| Substance | IP(eV) | | IP(eV) in the Presence of $H_2O$ | IP(eV) in the Presence of 2 $H_2O$ |
|---|---|---|---|---|
| | CCSD(T) | Experiment | CCSD(T) | CCSD(T) |
| Methanol | 10.81 | 10.84 | 9.63 | 9.17 |
| Acetone | 9.65 | 9.703 | 9.54 | — |
| n-Hexane | 9.96 | 10.13 | — | — |
| Isooctane | — | 9.89 | — | — |
| Methane | 12.66 | 12.61 | — | — |

We can notice that the CCSD(T) computed values are in good agreement with the experimental values, the discrepancy being 0.03, 0.05, 0.05 and 0.17 eV for methanol, acetone, methane, and n-hexane, respectively. It can be observed that the relative experimental values for acetone, n-hexane, isooctane and methane are in line with an electron transfer step as rate-determining ($k_{acetone} > k_{hexane} \approx k_{iso-octane} > k_{methane}$). On the contrary, the IP value of methanol (both the experimental and the computational one) is much higher than the expected ones: assuming the operation of an electron transfer mechanism, the above value should have been the lowest of the series.

We suggest that a possible justification of this apparent anomaly could be the presence of water molecules, always present at the $TiO_2$ surface, which can interact with polar molecules (through dipole-dipole interactions with acetone or dipole-dipole via hydrogen bond with methanol). Therefore, for methanol and acetone, we computed the IPs also in the presence of one water molecule. From the optimized geometries reported in Figure 8, we can notice that there is a strong interaction of the water molecule with ionized methanol.

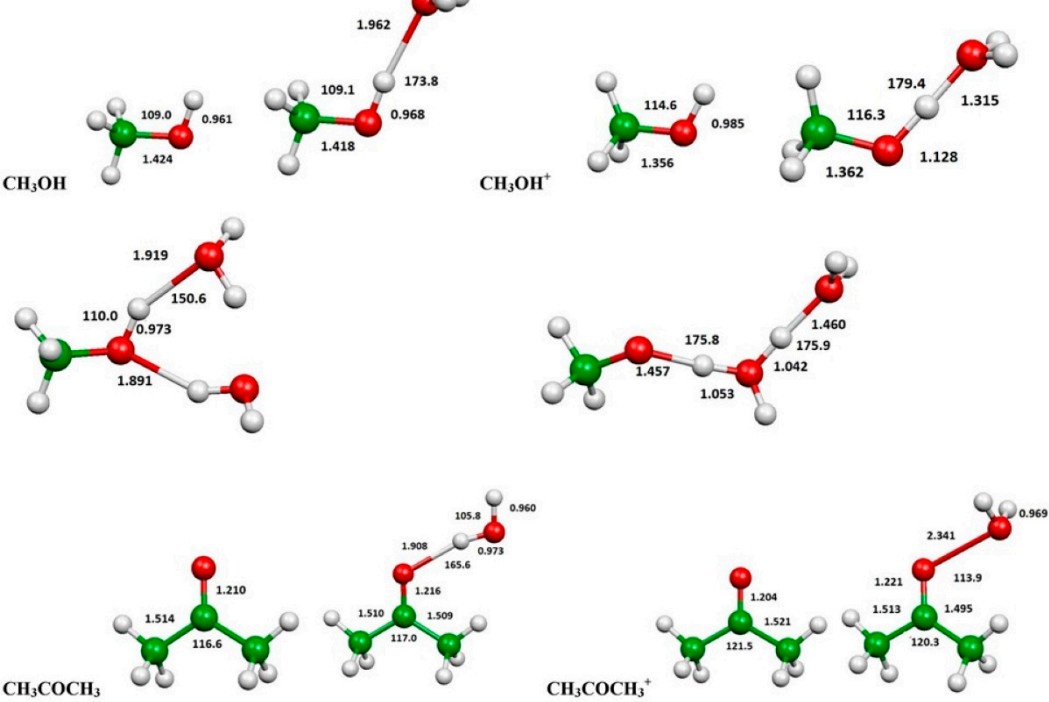

**Figure 8.** B3LYP optimized geometries (Å, and °) of the gas phase methanol and acetone neutrals and radical cations with and without surrounding water molecules (see text).

The $H_2O$—$HOCH_3$ distance, which is 1.962 Å in the neutral system, becomes 1.315 Å in the ionized complex; moreover, the methanol O—H bond length increases from 0.968 Å to 1.128 Å and the O—H—O angle, which is 173.8° in the neutral system, becomes 179.4° in the ionized one. The geometry of the complex $(CH_3OH^{..}OH_2)^+$·suggests, therefore, the presence of a hydrogen bond between methanol radical cation and water molecule. This is also confirmed by the Mulliken charges, reported in Figure 9 for methanol and acetone, which shows a charge transfer of 0.29 e from the water molecule to the ionized methanol molecule.

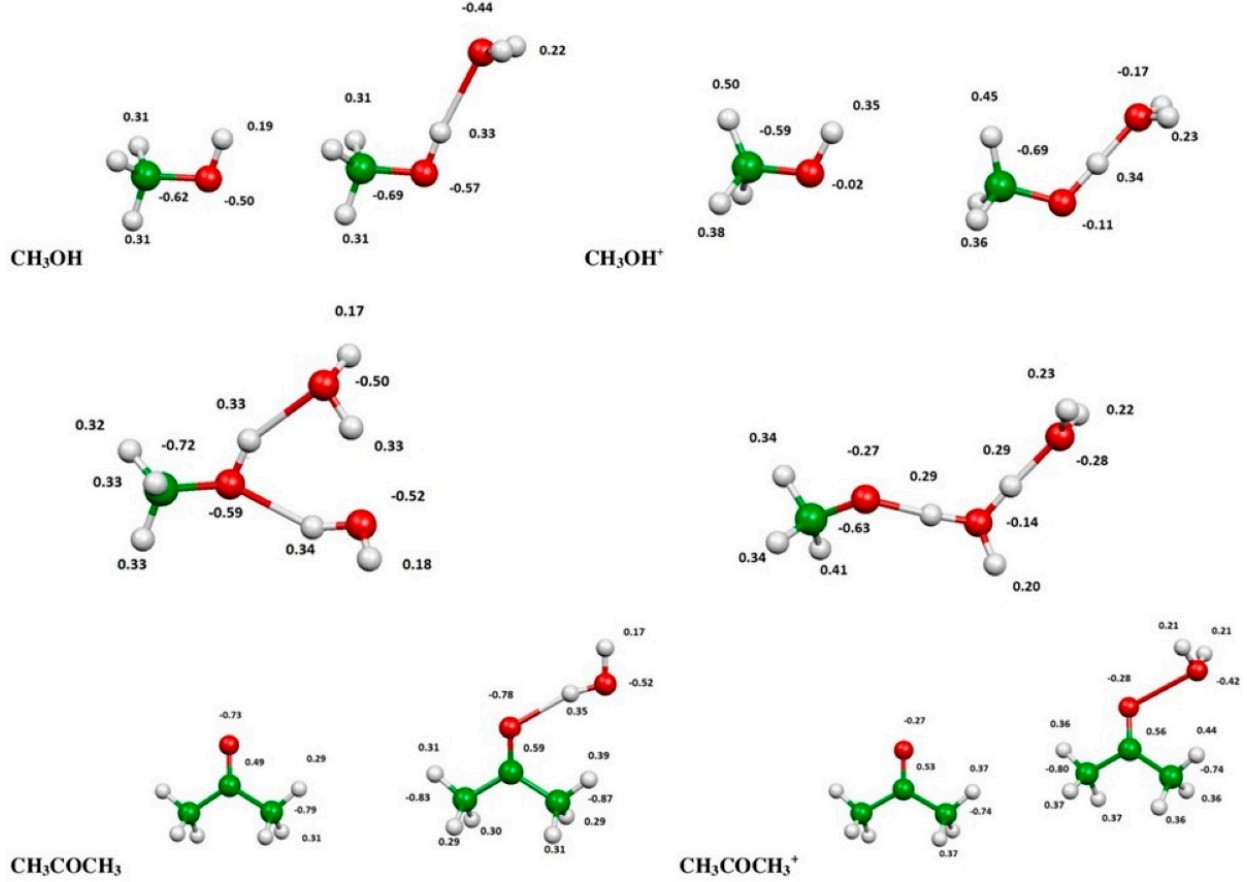

**Figure 9.** Calculated Mulliken charges for methanol and acetone neutrals and radical cations with and without surrounding water molecules.

This interaction decreases the energy of the ionized species to a greater extent with respect to the neutral species when one water molecule is present. The consequence is a smaller value of the IP due to the presence of one water molecule (9.63 eV instead of 10.81 eV, as we can see from Table 3). Since the influence of water on the ionization potential of methanol is very strong, we also investigated the effect of a second molecule of water. From the optimized geometries shown in Figure 8, we can notice that in the presence of a second molecule of water, we have a transfer of a proton from $CH_3OH^{+\bullet}$ to $H_2O$ and the system is best described as $CH_3O^{\bullet}$—$H_3O^+$—$H_2O$.

As we can see from Table 3, the ionization potential of methanol in the presence of two molecules of water is 9.17 eV (the lowest IP of the series); therefore, the IP decreases by 0.46 eV with respect to the presence of only one molecule of water. The total decrease of the IP of methanol, due to the presence of two water molecules, therefore, is 1.64 eV.

A water molecule does not seem to interact strongly with acetone (both neutral and ionized), as we can see from the optimized structures reported in Figure 8. Actually, we have a slightly stronger interaction with the neutral molecule (the $(CH_3)_2CO$—$HOH$ distance is 1.908 Å) than with the ionized one the $((CH_3)_2CO$—$OH_2)^+$·distance is 2.341 Å).

For this reason, the IP of acetone does not change significantly in the presence of one water molecule (9.54 eV instead of 9.65 eV). In addition, the Mulliken charges reported in Figure 9 show that the interaction of acetone with water is small, as we can notice from the lack of charge transfer from water to acetone.

## 3. Materials and Methods

### 3.1. Materials

Commercial samples of titanium(IV)tetraisopropoxide (TTIP, Sigma-Aldrich, St. Louis, MO, USA, 97%), $TiO_2$ (P25, Evonik Degussa, Essen, Germany), isopropanol (IP, Carlo Erba Reagents, Milano, Italy, 99.7%), acetone (Sigma-Aldrich, St. Louis, MO, USA, 99%), hexane (Sigma-Aldrich, St. Louis, MO, USA, 97%), isooctane (Honeywell—Riedel-de Haen, Charlotte, NC, USA, 99%), methanol (Carlo Erba Reagents, Milano, Italy, 99.9%), gas mixture ($O_2$, 20% and $N_2$, 80%, Air Liquide, Paris, France) were used. The gaseous mixture containing 600 ppm of $CH_4$ in an $N_2 + O_2$ mixture with a composition 80:20 was supplied by Air Liquide.

### 3.2. Synthesis of $TiO_2$ Powders

A complete description of the synthesis and characterization of the used $TiO_2$ powders is already published [20], and here is given only a brief summary of the procedure. By magnetic stirring at room temperature, 12 mL of TTIP were added to 500 mL of IP. Then, 2500 mL of Milli-Q water was added, and the final solution was stirred in a flask for 24 h. A white precipitate was obtained and then filtered and washed using Milli-Q water. The final white residue was first dried, and then, using a ceramic mortar, was ground. A thermal treatment on the obtained particles was performed for 1 h at 100, 200, 350, 500, 700, 850 and 1000 °C (a temperature gradient of 3 °C/min was used to reach each temperature value) [20].

### 3.3. Experimental Apparatus

The experiment exploited a cylindrical Pyrex glass static self-built prototype reactor already used in previous studies and described in detail elsewhere [20]. It was equipped with an optical window (Pyrex, thickness = 1 cm, diameter = 24 cm) and an o-ring with a metallic flange allowing a hermetic seal. Such a reactor was thermostated at 28 ± 3 °C by water cooling, and it has five connections: one is used for a fan; a second one is used for the insertion of the $N_2/O_2$ gaseous reaction mixture; additional connection allows the introduction of liquid reagents by a syringe. Then, first, electrical contact is for a gas sensor Delta Ohm (Padova, Italy) mod. HD37B17D (to monitor $CO_2$ concentration, relative humidity and temperature), while the second one is for a manometer Delta Ohm (Padova, Italy) mod. HD2304 (used to measure the pressure inside the reactor).

### 3.4. Photodegradation of Methane, Hexane, Isooctane, Acetone and Methanol in the Presence of $TiO_2$

Inside the reaction chamber, the synthetic $TiO_2$ powder (2 g) or TiO2 P25 (1 g) was inserted to completely cover the internal surface of a Petri plate (diameter = 10 cm) in support of Teflon. Then, the reactor was pumped and degassed at about 8 mbar with the Pyrex cap hermetically sealed and using a syringe ca. 600 ppm of the liquid reagent were inserted. Then, adding the $O_2/N_2$ gas mixture, a pressure of about 1.0 bar was set. In the case of methane, the commercial mixture was used at 1.0 bar pressure. After about 1 h that it was left in the dark, the reactor was irradiated by a 500 W high-pressure mercury lamp (Helios Italquartz at 300–580 nm). Irradiation of about 22 mW/m$^2$ on the photocatalyst was detected by a radiometer. The synthetic photocatalysts show light adsorption at wavelengths <580 nm. In the presence of the Pyrex cap of the glass reactor, this range becomes nearly 300–580 nm, a range in which all the synthesized $TiO_2$ samples adsorb the UV-vis radiation (see, for example, UV-vis DRS of samples prepared in the same way in. [26]).

### 3.5. Product Analysis

Since, in our case, the only products of the photodegradation reactions of the simple VOCs studied are carbon dioxide and water in stoichiometric quantities, we have quantitatively determined the $CO_2$ formed. A gas sensor Delta Ohm (Padova, Italy) mod. HD37B17D with an IR cell allowed the analysis of the produced $CO_2$. By a gas-chromatograph Varian 3900 (Agilent Technologies, Santa Clara, CA, USA), equipped with an FID detector and a CP-WAX 57CB capillary column (Agilent Technologies, Santa Clara, CA, USA), the quantitative determination of the unreacted hexane was performed.

### 3.6. Computational Evaluation of Ionization Potentials

In order to evaluate the ionization potentials of methanol, acetone, and n-hexane, we optimized the geometry of the neutral molecules, and the corresponding radical cations at the B3LYP [27,28] level of theory in conjunction with the correlation consistent valence polarized set aug-cc-pVTZ [29,30], as we have already done for different systems [31–33]. At the same level of theory, we have computed the harmonic vibrational frequencies. The energy of the stationary points was computed then at the higher level of calculation CCSD(T) [34–36] using the same basis set aug-cc-pVTZ. The CCSD(T) energies were corrected to 298.15 K by adding the zero-point energy correction and the thermal correction computed using the scaled harmonic vibrational frequencies evaluated at B3LYP/aug-cc-pVTZ level. The geometry of methanol and acetone was also optimized in the presence of one molecule of water in order to check the effect of this molecule on computed ionization potentials. For methanol, we also considered the effect of the presence of a second molecule of water on its ionization potential. All calculations were performed using Gaussian 09 [37], while the analysis of the vibrational frequencies was performed using Molekel [38,39].

## 4. Conclusions

In the present paper, we performed a jointed experimental and theoretical investigation on $TiO_2$ catalytic elimination of some VOCs environmental pollutants, pointing out new insights about the microscopic mechanism of the photocatalytic mineralization of five VOCs by $TiO_2$. By experiments, we were able to verify that the relative efficiency of acetone and hexane, as substrates having different polarity, for direct photolysis is maintained using $TiO_2$ powders with different properties (crystalline form, degree and size, surface area and pore volume). Furthermore, we have performed chemical kinetics experiments via a Langmuir–Hinshelwood approach for the determination of either K (the adsorption constant on $TiO_2$) and k (the mineralization rate constant) using five VOCs (methanol, acetone, isooctane, hexane, and methane). The recorded rate constant data showed that the nonpolar molecules (hexane and isooctane) are less reactive than the polar ones (acetone and methanol). These experimental data, together with a computational analysis of a high level of accuracy within our innovative approach, highlighted that the rate-determining step in the reaction mechanism of the investigated photosensitive oxidations should be an electron transfer step ($RH \rightarrow RH^+ + e^-$) instead of a possible competitive homolytic process ($RH \rightarrow R^\bullet + H^\bullet$). Such experimental evidence is corroborated by our performed theoretical calculations concerning geometry and energetic stability of both neutral and cationic species of the investigated VOCs. Using the B3LYP level of theory coupled with the correlation consistent valence polarized set aug-cc-pVTZ, we determined the ionization potentials for used VOCs in the presence of water molecules since water is always present at the $TiO_2$ surface strongly interacting via dipole-dipole interactions with polar molecules as acetone and methanol. The performed computational analysis demonstrated, in the case of methanol (the most reactive VOC among those investigated in the present work), the presence of a hydrogen bond between the radical cation and the surrounding water molecules strongly stabilizes such hydrated radical cation with respect to the case of isolated methanol. This causes a considerable decrease in the ionization potential of methanol when it is surrounded by two water molecules (in this case, the IP is ~1.64 eV

lower than the case of isolated methanol), justifying its stronger reactivity in photosensitive oxidations.

Finally, this work, with its mechanistic implications on basic chemistry, provides important evidence in the development of an innovative semiclassical model capable of fully describing the quantum state resolved reaction dynamics of prototype oxidation processes such as the chemi-ionization reactions [40,41].

**Author Contributions:** Conceptualization and methodology, S.F., C.R., M.R. and G.V.S. software and theoretical calculations, M.R.; formal analysis, C.R. and G.V.S.; investigation and data curation, M.B.; writing—original draft preparation, M.B., S.F., C.R., M.R. and G.V.S.; writing—review and editing, S.F., C.R., M.R. and G.V.S. All authors have read and agreed to the published version of the manuscript.

**Funding:** This research was funded by FONDAZIONE CASSA DI RISPARMIO DI PERUGIA, Italian MIUR and University of Perugia (Italy) within the program "Dipartimenti di Eccellenza 2018-2022".

**Conflicts of Interest:** The authors declare no conflict of interest. The funders had no role in the design of the study; in the collection, analyses, or interpretation of data; in the writing of the manuscript, or in the decision to publish the results.

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
