# Peer review of "Gas-Phase TiO2 Photosensitized Mineralization of Some VOCs: Mechanistic Suggestions through a Langmuir–Hinshelwood Kinetic Approach"

_catalysts, doi:10.3390/catal11010020_

Round 1
Reviewer 1 Report
All my comments I put in attached .pdf file.

Author Response
We thank you and both reviewers #1, and #2 for their comments and useful suggestions aimed to improve the quality of the manuscript ID Catalysts_1025159.
We made a revision of the manuscript, addressing all the comments and suggestions by the reviewers.
All made changes in the manuscript are highlighted in red color in its revised version (see new uploaded revised manuscript) and are listed below.
Responses to the Reviewer #1 comments:
Reviewer #1 - addressed point 1 at line 19: “Please add some numbers, what was the K and k range?”.
Author reply and made modifications: We agree with the reviewer #1 and we thank him for his suggestion. In order to address this point, we modified the sentence at line 19 as it follows: “Recorded K (in the range of 0.74×10-2 – 1.11×10-2 ppm-1) and k (in the range of 1.9 – 9.9 ppm min-1) values and performed theoretical calculations ….”;
Reviewer #1 – addressed point 2 at line 26: “Firstly long form, then shortcut.”.
Author reply and made modifications: we thank the reviewer for his suggestion and we modified the related sentence at line 26 of the revised manuscript as it follows: “The TiO2 photosensitized mineralization of most volatile organic compounds (VOCs) as…”;
Reviewer #1 – addressed point 3 at line 31: “photochemical or photocatalysis? photocatalysis is a surface reaction. this statement is obvious for photocatalysis.”.
Author reply and made modifications: we agree with the reviewer and we modified the related sentence at line 30 of the revised manuscript as it follows: “The photocatalytic process …”;
Reviewer #1 – addressed point 4 at line 38: “In my opinion "preadsorption" should be stated here.”.
Author reply and made modifications: the reviewer is right and at line 38 we changed the word “adsorption” with “preadsorption”;
Reviewer #1 – addressed point 5 at line 54: “why such concentration? how does it correspond to VOCs concentration in real samples?”.
Author reply and made modifications: in order to satisfy such a comment, at line 55 of the revised manuscript, we added the following sentence: “(the same experimental conditions used in a previous study on acetone [20] in order to make a comparison possible)”;
Reviewer #1 – addressed point 6 at lines 63-68: “This is Introduction, why the main conslusion is described? In Intro we need to state why the experiments were carried? what is the hypothesis? how can it be verified.”.
Author reply and made modifications: we agree with the reviewer and in order to satisfy such a comment, at lines 65-66 of the revised manuscript, we modified the sentence as it follows: “A careful analysis of the kinetic experimental evidences together with theoretical considerations was done (see below) in order to clarify if the rate determining step…”;
Reviewer #1 – addressed points 7-8 at lines 74-77: “The aim of the studies is not clear. Please stress how the photooxidation of hexane over different TiO2 is combined with photooxidation of other compounds on P25? I cannot find the stright information what for this was examined. Manuscript must be rearanged to stress its novelty. In present form is difficult to be understand. Self-citations. 4 in one sentence. ”.
Author reply and made modifications: probably, our sentence at lines 74-77 (“The expertise of our research group during last decade on the experimental and theoretical characterization of the chemical behaviour of ionic and radical species in gas phase [21,22] and in heterogeneous/homogeneous catalytic processes [23,24] has been used.”) is not clear and determines possible misunderstandings in the reader; in fact, the cited articles of our group are not directly related to the TiO2 studies: the sentence was intended to introduce the reader to the expertise of our research group that was used in the present manuscript, but this can be avoided since it can be considered as superfluous. In our opinion such a sentence is not necessary and we have decided to cancel it with the relative four self-citations [21-24];
The deleted four citations are the followings:
- Skouteris, D.; Balucani, N.; Faginas-Lago, N.; Falcinelli, S.; Rosi, M. Dimerization of methanimine and its charged species in the atmosphere of Titan and interstellar/cometary ice analogs. A&A 2015, 584, A76.
- Falcinelli, S.; Rosi, M.; Cavalli, S.; Pirani, F.; Vecchiocattivi, F. Stereoselectivity in Autoionization Reactions of Hydrogenated Molecules by Metastable Noble Gas Atoms: The Role of Electronic Couplings. Chem. Eur. J. 2016, 22, 12518-12526.
- Falcinelli, S.; Capriccioli, A.; Pirani, F.; Vecchiocattivi, F.; Stranges, S.; Martì, C.; Nicoziani, A.; Topini, E.; Laganà, A Methane production by CO2 hydrogenation reaction with and without solid phase catalysis. Fuel 2017, 209, 802-811.
- Falcinelli, S. Fuel production from waste CO2 using renewable energies. Catal. Today 2020, 348, 95-101.
Reviewer #1 – addressed point 9 at lines 93-94: “Here is a mixture of Experimental section and results. Please, make it clear and arrange properly.”.
Author reply and made modifications: we agree with the reviewer and we thank him for his careful check of the text; indeed, such a sentence (“The gas phase TiO2 photosensitized oxidation has been carried out in a thermostated static reactor (28±3°C, at constant humidity and oxygen concentration).”) is a repetition of what was stated in the “Experimental apparatus” section 3.3 (see lines 277-280) and must be deleted here as we have done in the revised version of the manuscript.
Reviewer 2 Report
Dear Authors,
The article presented for review concerns the issues related to photocatalysis on TiO2 oxide. This topic has been very widely discussed in various contexts for many years. What is true, this does not mean that all issues are already well known. However, to increase the attractiveness of this topic, I recommend making a graphic abstract. In this research topic, making a graphic abstract is almost a standard.
I believe that the theoretical introduction should be further expanded to include more detailed issues concerning the mechanism of the reaction taking place in this system. I believe that there is an imbalance between the amount of text describing what is already known about the Langmuir-Hinshelwood mechanism and what the authors want to present new.
There is no mention of a specific value of humidity and oxygen concentration in the text of the publication. I believe that the parameter related to the water content is important. Therefore, not listing these data but only briefly stating that the values of these parameters were kept at a constant level is insufficient.
Why in figure 1 the OY axis has such a strange signature? Wouldn't it be better to write ln(co/c)?
In the case of this work, I believe that all calculated values should have a specific measuring error range, as was done for figure 2.
The only thing that cannot be corrected in this publication is that the reagents used for the reaction were not of the highest purity, i.e. they contain too much water. In the case of the titanium precursor, all solvents used should have the lowest possible water content due to the rapid TTIP hydrolysis.
After reading the publication carefully, I have some grammar comments:
- In some words (very often repeated in the text) hyphen should be added to make the meaning of the word correct. Lines: 54, 64, 84, 93, 190, 193, 195, 197, 198, 217, 294, 316, 335.
- There are a few words in the text that should have an amended or added article. Lines: 76, 94, 141, 192, 197, 232, 242, 277 (for example an hydrogen?)
- I think you can consider changing the verb form for subject-verb agreement. Lines: 127, 145.
- In some words, the impression is that there are errors resulting from incorrect spelling. Lines: 68, 270, 297, 299, 330.
- The words 'exactly', 'contribute', 'that' doesn't seem to best fit this context. Lines 151, 190, 348.
- The words 'evidence' and "Teflon' are uncountable nouns, therefore the plural form is incorrect... Lines: 63, 289...
- Consider adding a comma after the introductory phase. Lines: 132, 207, 250, 320.
- Consider adding or replacing a preposition in some words. Lines: 178, 221.
- The verb 'obtain'... is usually in the gerund form when following the word allow. Lines: 96, 165.
Yours faithfully,
Reviewer
Author Response
We thank you and both reviewers #1, and #2 for their comments and useful suggestions aimed to improve the quality of the manuscript ID Catalysts_1025159.
We made a revision of the manuscript, addressing all the comments and suggestions by the reviewers.
All made changes in the manuscript are highlighted in red color in its revised version (see new uploaded revised manuscript) and are listed below.
Responses to the Reviewer #2 comments:
Reviewer #2 - addressed point 1: “…to increase the attractiveness of this topic, I recommend making a graphic abstract. In this research topic, making a graphic abstract is almost a standard.”.
Author reply and made modifications: We thank the reviewer #2 for his suggestion and we agree with him. For such a reason we uploaded the following graphical abstract in our submitted revision:
Reviewer #2 - addressed point 2: “I believe that the theoretical introduction should be further expanded to include more detailed issues concerning the mechanism of the reaction taking place in this system. I believe that there is an imbalance between the amount of text describing what is already known about the Langmuir-Hinshelwood mechanism and what the authors want to present new.”.
Author reply and made modifications: We thank the reviewer for his comment, since it allows us to clarify a possible misunderstanding that the reader might run into. For such a reason we added the following sentences at line 79 of the revised manuscript: “To our knowledge, there are no mechanistic studies concerning photocatalytic TiO2 mineralization of VOCs in gaseous phase in the literature. The present work wants to make a contribution in this sense but does not claim to carry out a study of the complex reaction mechanism of the investigated processes but only to point out the characteristics of the rate determining step using the mathematical LH approach.”.
Reviewer #2 - addressed point 3: “There is no mention of a specific value of humidity and oxygen concentration in the text of the publication. I believe that the parameter related to the water content is important. Therefore, not listing these data but only briefly stating that the values of these parameters were kept at a constant level is insufficient.”.
Author reply and made modifications: For this point we disagree with the reviewer since the manuscript already reports the relative humidity and oxygen values he requested: see at lines 120, 138, and 263 of the revised manuscript.
Reviewer #2 - addressed point 4: “Why in figure 1 the OY axis has such a strange signature? Wouldn't it be better to write ln(co/c)?.”.
Author reply and made modifications: According to the request of the reviewer, we modified the Fig.1 inserting the following new label for the y axis: “ln (C0/C) = ln [100/(100 − %CO2)]”.
Reviewer #2 - addressed point 5: “In the case of this work, I believe that all calculated values should have a specific measuring error range, as was done for figure 2.”.
Author reply and made modifications: We thank the reviewer and, according to his request, we added in the caption of the Figures 1, 3-7 the following sentence: “The experimental error is ≤ 10%.”.
Reviewer #2 - addressed point 6: “The only thing that cannot be corrected in this publication is that the reagents used for the reaction were not of the highest purity, i.e. they contain too much water. In the case of the titanium precursor, all solvents used should have the lowest possible water content due to the rapid TTIP hydrolysis.”.
Author reply and made modifications: In this case, we do not understand the reviewer’s comment above. In the presented study, we used all reagents in their highest grade of purity commercially available. The synthetic procedure is described in section 3.2, and is well-known and fully described in ref. [21] and references therein”.
Reviewer #2 – grammar comments: “After reading the publication carefully, I have some grammar comments:
In some words (very often repeated in the text) hyphen should be added to make the meaning of the word correct. Lines: 54, 64, 84, 93, 190, 193, 195, 197, 198, 217, 294, 316, 335.
There are a few words in the text that should have an amended or added article. Lines: 76, 94, 141, 192, 197, 232, 242, 277 (for example an hydrogen?)
I think you can consider changing the verb form for subject-verb agreement. Lines: 127, 145.
In some words, the impression is that there are errors resulting from incorrect spelling. Lines: 68, 270, 297, 299, 330.
The words 'exactly', 'contribute', 'that' doesn't seem to best fit this context. Lines 151, 190, 348.
The words 'evidence' and "Teflon' are uncountable nouns, therefore the plural form is incorrect... Lines: 63, 289...
Consider adding a comma after the introductory phase. Lines: 132, 207, 250, 320.
Consider adding or replacing a preposition in some words. Lines: 178, 221.
The verb 'obtain'... is usually in the gerund form when following the word allow. Lines: 96, 165.”.
Author reply and made modifications: We thank the reviewer for its scrupulous and useful reading with the suggested grammatical corrections. We have accepted all his indications by correcting all the points mentioned above in the revised text.
Round 2
Reviewer 1 Report
Authors have answered all reviewers comments snd questions. Now I recomend this manuscript ro be published in Catalysis.
Reviewer 2 Report
Dear Authors,
Compared to the previous version, I do notice a significant improvement, both in terms of content and grammar. The current version does not raise so many objections in me. But I still maintain the issue of purity of the reagents. I have personally checked the reagent labels that I have in my lab and additionally have looked through the websites. I think that in the long term perspective, you should purchase better quality reagents. Aldrich is not always the best solution.
Yours faithfully,
Reviewer